# Post-pandemic face-to-face learning environments for undergraduate students: A scoping review protocol

Karim Garzon Diaz[1]*, Angy Villamil Duarte[2], Sandra Velasco Forero[3], Sandra Herrera Jacobo[4]

1 Occupational Therapy Program- School of Medicine and Health Sciences, Universidad del Rosario, Bogotá, Colombia, 2 Medicine Program- School of Medicine and Health Sciences, Universidad del Rosario, Bogotá, Colombia, 3 Physiotherapy Program- School of Medicine and Health Sciences, Universidad del Rosario, Bogotá, Colombia, 4 Resource Center for Learning and Research, Universidad del Rosario, Bogotá, Colombia

☯ These authors contributed equally to this work.

* Karim.garzon@urosario.edu.co

**Data Availability Statement:** All relevant data are available at: https://doi.org/10.34848/GTLIUS.

**Funding:** The author(s) received no specific funding for this work.

## Abstract

### Objective

This scoping review explores the existing literature related to post-pandemic face-to-face learning environments for undergraduate students following their participation in virtual classes during the Covid-19 pandemic. The secondary objectives are focused to identify the cognitive; emotional, or communicative demands that characterize students; changes in pedagogical strategies; and primary recommendations regarding post-pandemic face-to-face education in the context of Covid-19.

## Introduction

The global emergency created by the presence of COVID-19 has led to significant change in the daily lives of people worldwide, ranging from lockdowns to the proliferation of virtual channels for social interaction and learning. Interest in its effects remains relevant in various fields including social, economic, health, technological, and educational studies. The transition back to face-to-face studies in university settings requires new adjustment processes for both students and teachers, necessitating pedagogical transformations and addressing cognitive, emotional, communicative, and habituation demands. Studies on the return to face-to-face studies in university settings are relatively scarce and warrant in-depth research.

## Inclusion criteria

This review will include studies completed and published in calendar years 2022 and 2024 that involve post-pandemic face-to-face learning environments for undergraduate students from any geographical area.

**Competing interests:** The authors have declared that no competing interests exist.

## Methodology

This scoping review will follow the JBI methodology for conducting scoping reviews. The following databases will be used: Academic Search Complete, Inter-American Development Bank, CLACSO, and The UNESCO Institute for Higher Education in Latin America and the Caribbean (IESALC), Dimensions, DOAB, DOAJ, ERIC, LILACS, Psicodoc, Redalyc, Redib, Scielo, Scopus, Dialnet, Web Of Science, Latindex, Google Scholar, SocArXiv. The search will aim to locate publications without language restrictions from any geographic location, including peer-reviewed articles, grey literature, preprints, technical notes, and policy reports. Three independent reviewers will screen, retrieve and review full-text studies and extract data. Consensus will be sought in the event of disagreement. The search results will be presented in the PRISMA-ScR flowchart. A narrative summary will also be included (Tricco AC 2018).

## Trial registration

**Registration of systematic reviews:** Open Science Framework (https://doi.org/10.17605/OSF.IO/6P9QB)

## Introduction

Interest in studying the impact of the COVID-19 pandemic has gradually emerged from the health sector, but also from political, economic, cultural, ecological, and educational perspectives, among others. Impacts that continue to be recognized for the influence of the pandemic in transforming the daily lives of people and the territories they inhabit.

The concept of "post-pandemic as defined by COVID-19" can refer to the period following the acute phase of the pandemic, as well as allude to significant changes in the experiences and lessons learned during the pandemic.

Meanwhile, Stiglitz [1] addresses the "concept of post-pandemic reconstruction" as the process of rebuilding and recovering from the consequences of the pandemic, as it has highlighted disparities in work organization and regional inequalities [2], which may include changes in the way we work, interact, and address issues related to education, health, and ecology, among others.

Moreover, the post-pandemic period has been understood as an event that has changed the course of people's lives and, consequently, has a direct impact on education, especially higher education, as it faces challenges in managing it from a comprehensive perspective [3]. In this regard, UNESCO [4] has shown that public policies and strategies are still inadequate and partial and require an intersectoral and multi-actor approach.

In the field of education, the impending triggered by the COVID-19 pandemic brough about sudden changes in learning styles and environments, affecting pedagogical practices for both students and teachers. The rapid transition from face-to-face to virtual education during periods of isolation, often referred to as "emergency education" [5], presented numerous challenges. These challenges included emotional adjustments, cognitive demands, exhaustion, strained student-teacher relationships, difficulties in organizing independent work and maintaining focus, and exacerbation of inequalities due to limited access to technology or digital divides [5]. At the same time, new family roles have emerged to support students academically at home, physical spaces have been adapted for home-based learning, and teacher expectations and workloads have increased.

From a higher education perspective, the impact periods of confinement on the return to face-to-face attendance is becoming increasingly clear. This generation of students "face unprecedented challenges in adapting to university life" [6], while for professors, it means new ways of adapting and understanding the realities imposed by virtual education. This includes understanding students' behaviors and attitudes, which require new modes of interpretation that can directly or indirectly affect post-pandemic face-to-face learning environments.

Reflective, analytical, and descriptive research on the impact of the pandemic has permeated various social, economic, health, and technological contexts. In higher education, studies have focused on mental health issues, behavioral concerns, students well-being, and perspectives on holistic student development [7]. In addition, some authors have examined the technological adaptations made by educators, including the implementation of new instructional strategies that utilize technology. Recommendations have been for pedagogical and curricular adaptations in higher education that address theoretical, practical, and professional components [8].

This research poses new challenges regarding the impact of pedagogical models, the interpretation of student engagement in post-pandemic times, their interactions with peers and instructors, their independent study habits, and consequently, the reflection on teaching methodologies, learning environments, and their influence on institutional or governmental policies [8].

Although learning environments are intentional spaces designed to support the acquisition of competencies or learning outcomes, they are more than just physical and pedagogical domains [9]. These environments are influenced by relational and affective factors [10] such as trust, motivation, efficacy, well-being, and participation. Therefore, it is crucial to consider how the effects of virtual education are reflected or interpreted in post-pandemic face-to-face learning environments.

It is also important to examine how learners' behaviors are interpreted in these settings and how these behaviors [11], along with decision-making processes, shape the learning environments [12]. This includes aspects such as program design, definition of learning strategies, teacher assessment methods, as well as students' engagement, organization of independent work, interaction with peers, and contribution to class participation.

A preliminary search was conducted in the Dimensions and Redalyc databases, revealing a limited number of studies pertaining to the return to face-to-face teaching. The majority of these studies are articles indexed in education journals and categorized under the following themes: Education, educational systems, curriculum, pedagogy, educational policy, and health sciences.

In addition, validation was performed in the JBI database of evidence-based review Practical Evidence (Ovid) and PROSPERO databases using specific search criteria. However, the results yielded a minimal number of published studies as of June 15, 2024.

## Research question

What is known about post-pandemic (Covid-19) face-to-face learning environments for undergraduate students?

Research subquestions

1. What cognitive, emotional and communicative demands characterize undergraduate students face-to-face Learning during the post-pandemic period?

2. What changes have been made in pedagogical or teaching-learning strategies in face-to-face education after the pandemic?

3. What are the main recommendations in terms of guidelines, policies or programs related to face-to-face education during the pandemic?

### Inclusion criteria

**Participants.** This review considers studies that include the post-pandemic face-to-face learning environments of undergraduate students. The review excludes virtual or dual post-pandemic learning environments.

**Concept.** This review includes that examine or incorporate post-pandemic face-to-face learning environments, address cognitive and emotional demands, pedagogical adaptation, identified barriers to participation, and study habits.

**Context.** This review includes studies conducted in undergraduate programs from all disciplines and geographic areas.

**Types of sources.** This scoping review will consider qualitative, quantitative, mixed-methods, case studies, opinion pieces, technical notes, grey literature, preprints, and policy reports as well as protocols for systematic reviews or scoping reviews.

## Methods

This scoping review will be conducted according to the JBI methodology for scoping reviews [13] and will follow the Preferred Reporting Items for Systematic Review and Meta-Analysis extension for Scoping Review (PRISMA-ScR) guidelines. This scoping review is registered in the Open Science Framework (https://doi.org/10.17605/OSF.IO/6P9QB)

### Search strategy

The search strategy will be developed with the participation of an interprofessional team consisting of an occupational therapist, a medical physician, a physiotherapist, and a librarian. The objective of this review will be to identify primary studies published in indexed journals, reports from international organizations, repositories or portals of grey literature or preprints, published since 2022, and corresponding to any undergraduate field of study.

On the other hand, according to controlled term frequency analyses, there are many expressions associated with the post-pandemic period of Covid-19, such as post-pandemic, post-Covid-19, post-pandemic, post-COVID. Although this may be a limitation, we will attempt to include all these variations in the search strategies.

In accordance with the recommendation of the JBI Manual (2024), there will be no restrictions on the inclusion of language-based sources for the purpose of this study [14].

The databases used for keyword searches are: Academic Search Complete, Inter-American Development Bank, CLACSO, The UNESCO International Institute for Higher Education in Latin America and the Caribbean (IESALC), Dimensions, DOAB, DOAJ, ERIC, LILACS, Psicodoc, Redalyc, Redib, Scielo, Scopus, Dialnet, Web Of Science, Latindex, Google Scholar, SocArXiv.

A preliminary search was conducted using Dimensions and Redalyc, focusing on the topics listed S1 Appendix. The textual words found in titles and abstracts, along with any identified keywords from relevant records, are used as references. Similarly, the reference lists of documents meeting the inclusion criteria will be examined for additional bibliographic material. Documents published from the year 2022 until June 15, 2024, will be included.

### Study selection

After the search, all identified records will be compiled and uploaded to Zotero 6.0.26 and duplicates will be removed. After a pilot test (20 records), each of the titles and abstracts will be

screened by 3 independent reviewers for evaluation against the review inclusion criteria. Potentially relevant papers, including titles and abstracts will be retrieved in full and citation details will be imported into the JBI System for Unified Information Management, Assessment and Review (JBI SUMARI; JBI, Adelaide, Australia). The full text of selected citations is assessed in detail against the inclusion criteria by 3 independent reviewers. Reasons for exclusion of full-text articles that did not meet the inclusion criteria will be recorded and reported in the scoping review. In case of disagreement, consensus will be sought. The search results will be comprehensively reported in the scoping review.

The search results will be fully reported in the final scoping review and presented in a Preferred Reporting Items for Systematic Reviews and Meta-Analyses for Scoping Reviews (PRISMA-ScR) flowchart [15].

## Data extraction

Data will be extracted from the articles included in the scoping review by three independent reviewers using the Scoping Review Protocol template provided by JBI [15]. The extracted data will include specific details about the participants, concept, context, study methods, and key findings relevant to the review question(s) (See S2 Appendix) [16]. The draft data extraction tool will be modified and revised as necessary during the data extraction process for each record included. The modifications will be detailed in the full scoping review. Any disagreements that arise between reviewers will be resolved by consensus.

## Data analysis and presentation

The extracted data will be presented in a graphical or tabular format according to the objectives and questions of this scoping review. Tables and graphs will include information on authors, title, year of publication, source/journal, concept, population, study methods, context and setting, and key findings related to the study of post-pandemic face-to-face learning environments for undergraduate students. A narrative summary will accompany the tabular and graphical results, describing how the findings relate to the review objectives and questions. Findings will be discussed in terms of practice and research. All authors will determine how the results are presented.

## Ethical considerations

This study was approved by the Research Ethics Committee of the Universidad del Rosario (Registration No. DVO 005 2450 CV 1789/2023

## Discussion

The data from this scoping review are expected to guide research on the impact of face-to-face learning environments on undergraduate students during the pandemic, providing relevant information for policymakers and higher education institutions in general. This includes understanding and adapting students and faculty and providing opportunities for scholarly research in this area.

The evaluation of the documents to be reviewed will be carried out without language restrictions. This will allow a closer approximation of the evidence and a diversification of sources.

## Supporting information

**S1 Checklist. Preferred Reporting Items for Systematic reviews and Meta-Analyses extension for Scoping Reviews (PRISMA-ScR) checklist.**
(DOCX)

**S1 Appendix. Search strategy.**
(DOCX)

**S2 Appendix. Data extraction instrument.**
(DOCX)

# Acknowledgments

To Dr. Andrés Isaza Restrepo, Professor of the School of Medicine and Health Sciences of the University, for acting as overall reviewers of the text of this protocol.

# Author Contributions

**Conceptualization:** Karim Garzon Diaz.

**Formal analysis:** Karim Garzon Diaz, Angy Villamil Duarte, Sandra Velasco Forero.

**Investigation:** Karim Garzon Diaz, Angy Villamil Duarte, Sandra Velasco Forero.

**Methodology:** Karim Garzon Diaz, Angy Villamil Duarte, Sandra Velasco Forero, Sandra Herrera Jacobo.

**Writing – original draft:** Karim Garzon Diaz, Angy Villamil Duarte, Sandra Velasco Forero.

**Writing – review & editing:** Karim Garzon Diaz, Angy Villamil Duarte, Sandra Velasco Forero, Sandra Herrera Jacobo.

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
