## [Decision Letter · Decision Letter 0]

22 Feb 2024

PONE-D-24-02071Post-pandemic face-to-face learning environments in undergraduate health sciences students: a scoping review protocolPLOS ONE

Dear Dr. Garzón Díaz,

Thank you for submitting your manuscript to PLOS ONE. After careful consideration, we feel that it has merit but does not fully meet PLOS ONE’s publication criteria as it currently stands. Therefore, we invite you to submit a revised version of the manuscript that addresses the points raised during the review process. Please submit your revised manuscript by Apr 07 2024 11:59PM. If you will need more time than this to complete your revisions, please reply to this message or contact the journal office at plosone@plos.org. Please include the following items when submitting your revised manuscript:A rebuttal letter that responds to each point raised by the academic editor and reviewer(s). You should upload this letter as a separate file labeled 'Response to Reviewers'.A marked-up copy of your manuscript that highlights changes made to the original version. You should upload this as a separate file labeled 'Revised Manuscript with Track Changes'.An unmarked version of your revised paper without tracked changes. You should upload this as a separate file labeled 'Manuscript'.

We look forward to receiving your revised manuscript.

Kind regards,

Alejandro Botero Carvajal, MD

Academic Editor

PLOS ONE

Journal Requirements:

- DOI: 10.11124/JBIES-22-00147

In your revision ensure you cite all your sources (including your own works), and quote or rephrase any duplicated text outside the methods section. Further consideration is dependent on these concerns being addressed.

3. Thank you for stating the following in your Competing Interests section: "NO authors have competing interests"

Reviewers' comments:

Reviewer's Responses to Questions

**Comments to the Author**

1. Does the manuscript provide a valid rationale for the proposed study, with clearly identified and justified research questions?

Reviewer #1: Yes

Reviewer #2: Partly

2. Is the protocol technically sound and planned in a manner that will lead to a meaningful outcome and allow testing the stated hypotheses?

Reviewer #1: Partly

Reviewer #2: Partly

3. Is the methodology feasible and described in sufficient detail to allow the work to be replicable?

Reviewer #1: Yes

Reviewer #2: Yes

4. Have the authors described where all data underlying the findings will be made available when the study is complete?

Reviewer #1: Yes

Reviewer #2: No

5. Is the manuscript presented in an intelligible fashion and written in standard English?

Reviewer #1: No

Reviewer #2: Yes

6. Review Comments to the Author

You may also provide optional suggestions and comments to authors that they might find helpful in planning their study.

Reviewer #1: The scoping review protocol includes important details of the proposed scoping review. The manuscript contains quite a few grammatical issues, which makes some sections difficult to understand. I have pointed out some of these areas, however, did not point out every issue and encourage the authors to carefully review the manuscript for grammatical issues and consider an editing service if needed.

Abstract

Lines 29-30: The authors may want to consider revising to something such as “This scoping review aims to explore the existing literature related to post pandemic learning environments for undergraduate health sciences students after participating in virtual classes during the COVID-19 pandemic.”

Lines 30-31: Is the secondary aim focused on post-pandemic teaching-learning processes or these processes in general?

Lines 37-38: This sentence seems to have some grammatical/punctuation issues.

Lines 39-40: The wording of this sentence is unclear. Is it trying to say that there is little research on this topic?

Line 44: Is the review looking at studies conducted in 2022 and 2023 or published in 2022 and 2023. It may also be helpful to specify whether this is calendar year or academic year if you’re referring to when the studies were conducted.

Lines 46-48: This sentence could benefit from rewording. It would be helpful to put the names of the databases after the term database rather than after the types of publications that will be included.

Lines 48-49: It would be helpful to mention the title/abstract screening as well.

Introduction

Line 58: Change COVID 19 to COVID-19

Line 107: Specify COVID-19 pandemic. The abstract mentions that this is after the return from virtual learning. That may be helpful to specify here as well.

Lines 107-110: The objectives in the abstract differ somewhat to what is listed here. The wording doesn’t need to be identical, but they should be getting across the same points.

Lines 112-131: The inclusion criteria would fit better under methods rather than introduction. Do the studies included need to specifically address the return to face-to-face learning environments or will studies be included regardless of whether they address this as long as they were conducted in face-to-face learning environments during the timeframe of interest. It would be helpful to specify the time period of the studies you’re looking at and indicate whether that time period was when the studies were conducted or published.

Methods

135-136: “in correspondence” doesn’t seem to be the best term to use here

Lines 144-145: Specify the specific databases that will be searched. These seemed to be mentioned in the abstract, but not the main text.

Lines 147: I suggest saying something like “identify additional relevant papers.”

Line 151: It would be helpful to specify if this is a pilot test of the title and abstract screening process.

Line 152: From what’s written in the text it sounds like each article will be reviewed by 3 independent reviewers. It would be helpful to add “each” before title and abstract if that’s the case.

Lines 152-153: It’s mentioned that SUMARI will be used for full-text review. Will this also be used for title and abstract screening or will a different program be used?

Lines 152-153 and 157-158: I’m a little unclear on the screening process. Do all 3 reviewers need to make the same decision in order for an article to move forward? How is a fourth reviewer going to break a tie between three reviewers – you can get two and two?

Line 168: Change “scope” to “scoping”

Discussion

The discussion section would benefit from additional details, including a discussion of the limitations of the protocol.

Search strategy: I encourage the authors to further refine the search strategy and present a draft search strategy for at least one electronic database as outlined in the PRISMA-P checklist the authors included rather than information on the limited preliminary search that only provides one term for each concept.

Reviewer #2: Thank you for inviting me to review this manuscript and provide an opinion.

The authors will examine the important issue of post-pandemic in-person learning environments for health students. To achieve this, they are proposing a scoping review, using the JBI methodology. The methodological approach is appropriate, taking into account the objectives and research question.

Overall, the protocol is well written. I followed the steps proposed by the JBI methodology and articles previously published by PLOS ONE. It was based on these two references that I prepared my opinion.

I suggest some adjustments to the protocol to ensure the reliability and reproducibility of the review.

I answered questions 1 and 2 as partly because the search strategy and data extraction needs to be clear, as I describe later.

The last paragraph of the introduction describes the preliminary search and is appropriate, as JBI recommends, “The first step is an initial limited search of at least two appropriate online databases relevant to the topic." However, the details of databases, gray literature sites, and reference list searches must be clearly described in the protocol method. This information appears in the abstract, but is not included in the protocol method. The information can be within the search strategy or you can create a specific topic, “Information Sources." See these protocols, for example:

Within the search strategy: https://journals.plos.org/plosone/article?id=10.1371/journal.pone.0269495

Topic “Information Sources”: https://journals.plos.org/plosone/article?id=10.1371/journal.pone.0296659

Regarding information sources, it needs to be clear what the databases and gray literature sources will be and, also, add the search on websites of organizations for references on the subject, as appears in the method in the last example mentioned.

The authors chose to collect only in the English and Spanish languages. Which is not recommended by the JBI Manual. The recommendation is that “The time frame (start and end dates) chosen for the search should be clearly justified and any language restrictions specified (e.g. “only sources of evidence published in English were considered for inclusion”)." The authors need to eliminate this criterion or justify your choice. If researchers only speak English and Spanish, this does not justify the exclusion of the others languages. By the way, as it is a post-pandemic study, it must establish and justify what will be considered “post-pandemic”. See: https://jbi-global-wiki.refined.site/space/MANUAL/4687620/11.3.7.1+Search+strategy

The method informs that the evaluation “will be screened by three independent reviewers." Although the manual allows this to be “is performed by two or more reviewers, independently," it is totally acceptable for the evaluation to be carried by two independent reviewers and the third to be the judge in cases of disagreement. This would make a fourth evaluator unnecessary to resolve disagreements. As the team is made up of four people, it is possible to form two pairs of evaluators, and someone from the other pair can be a judge in the case of disagreement. This can make document evaluation faster. This paragraph is just a suggestion.

The extraction instrument needs to be improved, incorporating details of the questions and inclusion criteria that will be collected. For example, the concept appears: “aspects related to cognitive and emotional demands, pedagogical adjustments, detected barriers to participation, and study habits." I suggest that each topic be collected separately; this will make it easier to organize the results. If they collect them all together, the authors will have trouble separating them later. All the data that will be collected must be described, both in the method and in the extraction instrument. This is not clear in the protocol. As it stands, it is unclear what data will be collected.

*In addition to enabling reproducibility, the clearly constructed extraction instrument will facilitate researchers' work. We developed a scope review, in the research group that I lead and left the instrument very open. When it came to organizing the results, it was much more difficult.

Reference 10 is wrong, the correct one is “Peters MDJ, Godfrey C, McInerney P, Munn Z, Tricco AC, Khalil, H. Chapter 11: Scoping Reviews (2020 version). Aromataris E, Munn Z, editors. JBI Manual for Evidence Synthesis. JBI; 2020. Available from https://synthesismanual.jbi.global . https://doi.org/10.46658/JBIMES-20-12

I suggest that the other references be reviewed; I didn't check them all.

7. PLOS authors have the option to publish the peer review history of their article (what does this mean?). If published, this will include your full peer review and any attached files.

Reviewer #1: No

Reviewer #2: **Yes: **Maristela Santini Martins

---

## [Author Response · Author response to Decision Letter 0]

8 Apr 2024

Thank you very much, we have addressed all the comments and are attaching the required files.

---

## [Decision Letter · Decision Letter 1]

16 May 2024

PONE-D-24-02071R1Post-pandemic face-to-face learning environments in university students: a scoping review protocolPLOS ONE

Dear Dr. Garzón Díaz,

Thank you for submitting your manuscript to PLOS ONE. After careful consideration, we feel that it has merit but does not fully meet PLOS ONE’s publication criteria as it currently stands. Therefore, we invite you to submit a revised version of the manuscript that addresses the points raised during the review process.

We look forward to receiving your revised manuscript.

Kind regards,

Alejandro Botero Carvajal, MD

Academic Editor

PLOS ONE

Journal Requirements:

Reviewers' comments:

Reviewer's Responses to Questions

**Comments to the Author**

1. Does the manuscript provide a valid rationale for the proposed study, with clearly identified and justified research questions?

Reviewer #1: Yes

Reviewer #2: Yes

2. Is the protocol technically sound and planned in a manner that will lead to a meaningful outcome and allow testing the stated hypotheses?

Reviewer #1: Partly

Reviewer #2: Partly

3. Is the methodology feasible and described in sufficient detail to allow the work to be replicable?

Reviewer #1: Yes

Reviewer #2: Yes

4. Have the authors described where all data underlying the findings will be made available when the study is complete?

Reviewer #1: Yes

Reviewer #2: Yes

5. Is the manuscript presented in an intelligible fashion and written in standard English?

Reviewer #1: No

Reviewer #2: Yes

6. Review Comments to the Author

You may also provide optional suggestions and comments to authors that they might find helpful in planning their study.

**Reviewer #1**: The authors have made helpful revisions to the manuscript. Below are some additional suggestions to help further strengthen the manuscript. The manuscript still contains some grammatical issues (I pointed out some but not all), so I encourage carefully reviewing for grammar.

There are discrepancies in the title. The information on the title page and version without highlights use the title “Post-pandemic face-to-face learning environments in university students: a scoping review protocol” and the version with the highlights uses “Post-pandemic face-to-face learning environments in undergraduate health sciences students: a scoping review protocol.”

Abstract

Lines 26-28: Since the first component includes commas, it would be beneficial to use semi-colons to separate each element.

Line: 44: It’s unclear what is meant by “designed for this purpose.” I suggest specifying this refers to conducting scoping reviews.

Lines 50-51: The wording of this sentence doesn’t make it clear that articles will be screened for eligibility criteria. It sounds like all articles retrieved will be included in the study.

Lines 51-52: This sentence is a little unclear and conflicts with lines 188-189, which state that conflicts will be resolved by consensus OR a fourth reviewer. The wording of this sentence makes it sound like a fourth reviewer will be brought in to moderate the consensus process among the three original reviewers.

Lines 65-66: The wording of this sentence is somewhat awkward.

Lines 70-71: There seems to be an “and” missing from “education, health, ecology”

Lines 95-98: These sentences seem to be missing references

Lines 129-130: Change “in-face to face learning” to “in face-to-face learning”

Line 131: Is this referring to changes compared with pre-pandemic?

Lines 164-166: This section could benefit from reframing. Different terms representing the concepts of interest is something should be considered in developing all search strategies and isn’t unique to this study. Including different terms used to search for different concepts is beneficial and including an appendix or table with the search terms used for the different concepts within the manuscript can be helpful. The post-pandemic period isn’t the only concept within the research question that has numerous terms associated with it and those should be described here as well.

Line 177: This sentence has grammatical issues.

Line 216: This includes grammatical issues

Lines 221-222: The phrase” This will allow, on one hand, to gain a closer approximation to the evidence and diversification of sources” is unclear.

Search strategy: The search strategy provided is a preliminary search and does not reflect the updates to the research question. It focuses on health sciences students. JBI recommends including a full search strategy for at least one electronic database as part of the protocol.

**Reviewer #2**: The manuscript presents a considerable evolution compared to the first version. But some final adjustments still need to be made, so as to avoid doubts for readers later.

The search strategy is presented in Appendix II, but it seems to me to be incomplete. The archive needs to present the complete strategy for databases, which differ in some aspects, for gray literature organizations and websites.

In the search strategy, separate what are the databases, organizations, and websites of the gray literature. This way, it will be more suitable for insertion in the PRISMA flow diagram.

Although the PLOS ONE journal itself publishes scoping review protocols that use only English as an inclusion criterion, I maintain the recommendation of not restricting languages for inclusion in the review. The 2024 version of the JBI manual strongly recommends that there be no language restrictions.

“Reviewers should include the languages that will be considered for inclusion in the review as well as the timeframe, with an appropriate and clear justification for choices. Our strong recommendation is that there are no restrictions on source inclusion by language unless there are clear reasons for language restrictions (such as for feasibility reasons).”.

https://jbi-global-wiki.refined.site/space/MANUAL/355862729/10.2.5+Search+Strategy

The justification that “as the team's linguistic capabilities and because we do not have translation services for other languages” is not reasonable. I clarify that, in my opinion, the issue of language restrictions was not considered in the minor revision recommendation and will not be considered in the next revision of this protocol, but I leave it on record here.

Regarding the selection of studies, the authors maintain that “each of the titles and abstracts will be screened by three independent reviewers for evaluation against the inclusion criteria for review” and add that “any disagreements that arise between reviewers at each stage of the screening process will be resolved by consensus or with a fourth reviewer." It makes no sense. An additional reader is required in the absence of consensus on document inclusion or exclusion. In the case of three reviewers, there will always be a consensus; the three can agree to include or exclude, or there will be two in favor or two against. Are authors considering consensus only when all three readers agree among themselves? I understand that the JBI Manual proposes that the third reviewer is precisely to resolve the lack of consensus.

https://jbi-global-wiki.refined.site/space/MANUAL/355862749/10.2.6+Source+of+evidence+selection

Although the authors cited The extracted data will include specific details about the “participants, concept, context, study methods, and key findings relevant to the review question(s) (see Appendix II)”. I did not find the Appendix in the document II.

7. PLOS authors have the option to publish the peer review history of their article (what does this mean?). If published, this will include your full peer review and any attached files.

Reviewer #1: No

Reviewer #2: **Yes: **Maristela Santini Martins

---

## [Author Response · Author response to Decision Letter 1]

27 Jun 2024

Bogota, June 24 - 2024

Dear

Editorial Manager

Plos One

Following the instructions received, we have reviewed the manuscript titled “Post-pandemic face-to-face learning environments for undergraduate: a scoping review protocol”. We are now submitting responses to each of the points raised by both the editor and the reviewers.

Comments Reviewer #1 Authors response

“There are discrepancies in the title. The information on the title page and version without highlights use the title “Post-pandemic face-to-face learning environments in university students: a scoping review protocol” and the version with the highlights uses “Post-pandemic face-to-face learning environments in undergraduate health sciences students: a scoping review protocol.”

 In the previous revision, we presented in the "Response to reviewers" letter (pages #2 and 3) our intention to reformulate the study's questions and sub-questions, and we argued for the adjustment of the title accordingly, thus:

“To enhance clarity, we decided to focus the study on "undergraduate students," regardless of their field of study, and to reformulate the study's general question, as well as to define three sub-questions. In this way, we hope to have improved the data extraction tool to facilitate the analysis and interpretation of the findings”.

On the other hand, in the “Revised Manuscript with Track Changes” file, we also note the adjustment to the title (highlighted in blue).

Abstract

Lines 26-28: Since the first component includes commas, it would be beneficial to use semi-colons to separate each element. Adjustments made. In the “Revised Manuscript with Track Changes” version, semi-colons were included.

Line: 44: It’s unclear what is meant by “designed for this purpose.” I suggest specifying this refers to conducting scoping reviews. Adjustments made, In the “Revised Manuscript with Track Changes” version: “This scoping review will follow the JBI methodology to conducting scoping reviews” 

Lines 50-51: The wording of this sentence doesn’t make it clear that articles will be screened for eligibility criteria. It sounds like all articles retrieved will be included in the study. The sentence was adjusted. In the “Revised Manuscript with Track Changes” was included “screen”, before “retrieve”.

Lines 51-52: This sentence is a little unclear and conflicts with lines 188-189, which state that conflicts will be resolved by consensus OR a fourth reviewer. The wording of this sentence makes it sound like a fourth reviewer will be brought in to moderate the consensus process among the three original reviewers. In the “Revised Manuscript with Track Changes”, the wording of lines 188-189 was adjusted to be consistent with lines 51-52.

The authors chose only 3 reviewers, and consensus will be used.

New Lines 54-55-202-206.

Lines 65-66: The wording of this sentence is somewhat awkward. In the “Revised Manuscript with Track Changes” the wording was improved.

New Line 71-72.

Lines 70-71: There seems to be an “and” missing from “education, health, ecology” In the “Revised Manuscript with Track Changes” the Word “and” was included.

New Line 79

Lines 95-98: These sentences seem to be missing references In the “Revised Manuscript with Track Changes”, the reference was included.

New Line 111

Lines 129-130: Change “in-face to face learning” to “in face-to-face learning” In the “Revised Manuscript with Track Changes”, the hyphen was eliminated.

New Line 129

Line 131: Is this referring to changes compared with pre-pandemic? Yes, it is referring to changes compared with pre-pandemic.

Lines 164-166: This section could benefit from reframing. Different terms representing the concepts of interest is something should be considered in developing all search strategies and isn’t unique to this study. Including different terms used to search for different concepts is beneficial and including an appendix or table with the search terms used for the different concepts within the manuscript can be helpful. The post-pandemic period isn’t the only concept within the research question that has numerous terms associated with it and those should be described here as well. Within the protocol, we make reference to these controlled terms, anticipating that following the content analysis of the reviews, we may uncover evidence enabling the structuring of a specific appendix

Line 177: This sentence has grammatical issues. In the “Revised Manuscript with Track Changes”, the grammatical issues were improved.

New Lines 195-196

Line 216: This includes grammatical issues In the “Revised Manuscript with Track Changes”, the grammatical issues were improved.

New Lines 237-238

Lines 221-222: The phrase” This will allow, on one hand, to gain a closer approximation to the evidence and diversification of sources” is unclear. According to this observation and considering the comment from reviewer #2 to not restrict the search to English and Spanish, we decided to adopt this recommendation.

New Lines 245-246

Search strategy: The search strategy provided is a preliminary search and does not reflect the updates to the research question. It focuses on health sciences students. JBI recommends including a full search strategy for at least one electronic database as part of the protocol. The search strategy is updated based on the controlled terms.

Comments Reviewer #2 Authors response

The search strategy is presented in Appendix II, but it seems to me to be incomplete. The archive needs to present the complete strategy for databases, which differ in some aspects, for gray literature organizations and websites.

In the search strategy, separate what are the databases, organizations, and websites of the gray literature. This way, it will be more suitable for insertion in the PRISMA flow diagram.

 The search strategy presented in Appendix II was adjusted to differentiate between databases and other search sources such as grey literature organizations and websites, among others.

Although the PLOS ONE journal itself publishes scoping review protocols that use only English as an inclusion criterion, I maintain the recommendation of not restricting languages for inclusion in the review. The 2024 version of the JBI manual strongly recommends that there be no language restrictions.

“Reviewers should include the languages that will be considered for inclusion in the review as well as the timeframe, with an appropriate and clear justification for choices. Our strong recommendation is that there are no restrictions on source inclusion by language unless there are clear reasons for language restrictions (such as for feasibility reasons)”. 

The justification that “as the team's linguistic capabilities and because we do not have translation services for other languages” is not reasonable. I clarify that, in my opinion, the issue of language restrictions was not considered in the minor revision recommendation and will not be considered in the next revision of this protocol, but I leave it on record here. In accordance with this reviewer's observation and the 2024 version of the JBI manual, the author team will adopt the no language restrictions. This adjustment is made in the new version of the manuscript (“Revised Manuscript with Track Changes”).

A new reference was included in this section, citing the new JBI manual.

Additionally, the review timeframe was also adjusted to June 05, 2024

Regarding the selection of studies, the authors maintain that “each of the titles and abstracts will be screened by three independent reviewers for evaluation against the inclusion criteria for review” and add that “any disagreements that arise between reviewers at each stage of the screening process will be resolved by consensus or with a fourth reviewer." It makes no sense. An additional reader is required in the absence of consensus on document inclusion or exclusion. In the case of three reviewers, there will always be a consensus; the three can agree to include or exclude, or there will be two in favor or two against. Are authors considering consensus only when all three readers agree among themselves? I understand that the JBI Manual proposes that the third reviewer is precisely to resolve the lack of consensus. Following the recommendations, the involvement of a fourth reviewer was removed from the manuscript, and it is maintained that in the event of any discrepancies, consensus will be utilized.

This adjustment is made in the new version of the manuscript (“Revised Manuscript with Track Changes”).

Although the authors cited the extracted data will include specific details about the “participants, concept, context, study methods, and key findings relevant to the review question(s) (see Appendix II)”. I did not find the Appendix in the document II. The following supporting documents are attached: Checklist, Search strategy, and Data extraction instrument.

We appreciate the work done by the publisher and the reviewers; it has allowed us to gain precision regarding the scope of the manuscript. We hope the second version meets the publication requirements. We also conducted a comprehensive review of the manuscript's grammar and adjusted for the new version.

Karim Garzón-Díaz

Corresponding author.

---

## [Decision Letter · Decision Letter 2]

12 Aug 2024

Post-pandemic face-to-face learning environments for undergraduate students: a scoping review protocol

PONE-D-24-02071R2

Dear Dr. Garzón Díaz,

We’re pleased to inform you that your manuscript has been judged scientifically suitable for publication and will be formally accepted for publication once it meets all outstanding technical requirements.

Kind regards,

Alejandro Botero Carvajal, MD

Academic Editor

PLOS ONE

Additional Editor Comments (optional):

Reviewers' comments:

Reviewer's Responses to Questions

**Comments to the Author**

1. Does the manuscript provide a valid rationale for the proposed study, with clearly identified and justified research questions?

Reviewer #2: Yes

2. Is the protocol technically sound and planned in a manner that will lead to a meaningful outcome and allow testing the stated hypotheses?

Reviewer #2: Yes

3. Is the methodology feasible and described in sufficient detail to allow the work to be replicable?

Reviewer #2: Yes

4. Have the authors described where all data underlying the findings will be made available when the study is complete?

Reviewer #2: Yes

5. Is the manuscript presented in an intelligible fashion and written in standard English?

Reviewer #2: Yes

6. Review Comments to the Author

You may also provide optional suggestions and comments to authors that they might find helpful in planning their study.

Reviewer #2: The authors adhered to the recommendations outlined in the opinion. For publication, they need to replace the PRISMA Checklist included in the annex with the specific Checklist for Scoping Reviews, which is available at: https://www.prisma-statement.org/scoping

7. PLOS authors have the option to publish the peer review history of their article (what does this mean?). If published, this will include your full peer review and any attached files.

Reviewer #2: **Yes: **Maristela Santini Martins

---

## [Editor Report · Acceptance letter]

12 Sep 2024

PONE-D-24-02071R2 

PLOS ONE

Dear Dr. Garzón Díaz, 

I'm pleased to inform you that your manuscript has been deemed suitable for publication in PLOS ONE. Congratulations! Your manuscript is now being handed over to our production team.

Kind regards, 

on behalf of

Dr. Alejandro Botero Carvajal 

Academic Editor

PLOS ONE